# Co-Production of Poly(3-hydroxybutyrate) and Gluconic Acid from Glucose by *Halomonas elongata*

**DOI:** 10.3390/bioengineering10060643

**Published:** 2023-05-25

**Authors:** Tânia Leandro, M. Conceição Oliveira, M. Manuela R. da Fonseca, M. Teresa Cesário

**Affiliations:** 1IBB-Institute for Bioengineering and Biosciences, Bioengineering Department, Instituto Superior Técnico, Universidade de Lisboa, Av. Rovisco Pais, 1049-001 Lisboa, Portugal; tania.leandro@tecnico.ulisboa.pt (T.L.); manuela.fonseca@tecnico.ulisboa.pt (M.M.R.d.F.); 2Associate Laboratory i4HB, Institute for Health and Bioeconomy, Instituto Superior Técnico, Universidade de Lisboa, 1049-001 Lisbon, Portugal; 3Centro de Química Estrutural, Institute of Molecular Sciences, Instituto Superior Técnico, Universidade de Lisboa, Av. Rovisco Pais, 1049-001 Lisboa, Portugal; conceicao.oliveira@tecnico.ulisboa.pt

**Keywords:** halophiles, *Halomonas elongata*, polyhydroxyalkanoates, gluconic acid

## Abstract

Polyhydroxyalkanoates (PHA) are biopolyesters regarded as an attractive alternative to petroleum-derived plastics. Nitrogen limitation and phosphate limitation in glucose cultivations were evaluated for poly(3-hydroxybutyrate) (P(3HB)) production by *Halomonas elongata* 1H9^T^, a moderate halophilic strain. Co-production of P(3HB) and gluconic acid was observed in fed-batch glucose cultivations under nitrogen limiting conditions. A maximum P(3HB) accumulation of 53.0% (*w*/*w*) and a maximum co-production of 133 g/L of gluconic acid were attained. Fed-batch glucose cultivation under phosphate limiting conditions resulted in a P(3HB) accumulation of only 33.3% (*w*/*w*) and no gluconic acid production. As gluconic acid is a valuable organic acid with extensive applications in several industries, this work presents an interesting approach for the future development of an industrial process aiming at the co-production of an intracellular biopolymer, P(3HB), and a value-added extracellular product, gluconic acid.

## 1. Introduction

Polyhydroxyalkanoates (PHAs) are microbial-produced thermoplastics that are also biodegradable and biocompatible. These polyesters are great alternatives to synthetic plastics, but the major challenge for PHA commercialization is its high production cost. The quest to produce PHAs in a cost-effective way has directed the attention to halotolerant and halophilic PHA-accumulating marine bacteria, as they can be cultivated at high salt concentrations under conditions that discourage the growth of contaminants. This allows for the possibility for designing open non-sterile cultivations and thus reducing operational costs. While halotolerant microorganisms can grow either with or without salt, halophiles require NaCl to grow, and they are classified as slight (1–5% NaCl), moderate (5–20% NaCl), and extreme halophiles (20–30%) [1].

*Halomonas* are halophilic bacteria belonging to the *Gammaproteobacteria* class, and some members were shown to have the capacity to produce considerable amounts of valuable PHA polymers [2,3] under nutrient-limiting conditions and in the presence of excess carbon [4,5,6]. Species such as *H. boliviensis* [7], *H. elongata* [6], *H. bluephagenesis* TD1 [8,9], *H. halophila* [10], *H. campaniensis* LS21 [11], and *H. venusta* [12] were described as effective PHA-producers. The most common type of PHA produced from mono- or disaccharides by *Halomonas* species is the homopolymer poly-3-hydroxybutyrate P3HB [3]. So far, the only *Halomonas* strain reported to produce the co-polymer P3HB-3HV using solely glucose is *H. campaniensis* LS21 [11].

In a previous work and aiming at further reducing PHA production costs, in particular the carbon source-related costs, hydrolysates of cellulose-rich residues of the red seaweed *Gelidium corneum* after agar extraction were used as a sugar source [13]. *Halomonas boliviensis* was the selected PHA-producer due to its reported ability to accumulate polymer between 50–88% *w*/*w* of its dry-weight [14] and for being able to grow on galactose, one of the sugars present in *Gelidium* hydrolysates. A P(3HB) content of 70% (*w*/*w*) was attained in fed-batch cultivations under phosphorous limitation using glucose. Comparable P(3HB) titers in assays using glucose or seaweed hydrolysates as C-source were also demonstrated [13].

Hydrolysates of the seaweed *Ulva lactuca* are currently being assessed as a C-source for PHA production. *Halomonas elongata* 1H9^T^, a halotolerant bacterium, was selected because of its versatility to metabolize different simple sugars, including those released during the hydrolysis of the polysaccharide fraction of *U. lactuca* (results to be published). This strain can grow and accumulate P3HB up to 50% of its dry weight under 0.6 to 1.4 M NaCl concentrations [1,6]. *H. elongata* 1H9^T^ has mainly been studied as an efficient commercial producer of ectoine, a compatible solute, which is accumulated internally as a strategy to protect the cell when cultivation happens under conditions of high osmotic stress [15]. With regard to PHA production, work has been developed with this strain [4,6]. In the present study, the best conditions to improve P(3HB) productivity by *H. elongata* 1H9^T^ were assessed in fed-batch cultivations using glucose under limitation of nitrogen or phosphorous.

Similarly to what was observed with *H. boliviensis* [13], gluconic acid was detected in *H. elongata* cultivations growing on glucose. Other members of *Halomonas* spp. have been reported to secrete and accumulate organic acids, such as oxalacetic and pyruvic acids, into the cultivation medium [16,17]. Gluconic acid production was also observed in *Chromohalobacter salexigens* (formerly *Halomonas elongata* 1H11 (DSM 3043) cultures [18]. The ability of *Halomonas elongata* 1H9^T^ to synthesize gluconic acid is here reported for the first time.

Gluconic acid is an organic acid with multiple uses, for example, as a food additive (E574) and in the construction industry as a retarder to concrete setting. Additionally, gluconate is used in therapeutics as an ion carrier. The concomitant production of P(3HB), an intracellular product, and of gluconic acid, which is excreted to the medium, might become an attractive integrated process in a bio-refinery context, as a valuable asset for decreasing the overall cost of P(3HB) production. The rationale for the co-production of PHA and extracellular value-added products (VAP) has been addressed by several authors and thoroughly reviewed recently by Li et al., 2017; Kumar and Kim, 2018; and Yadav et al., 2021 [19,20,21]. The range of VAP covers alcohol, hydrogen, ectoines, organic acids, amino acids (namely L-tryptophan), enzymes (alpha-amylase), exopolysaccharides, surfactants, and carotenoids. Among these, PHA co-production with ethanol or with hydrogen deserve special attention because of the importance of these biofuels in the green energy market [22,23]. Further, the co-production of PHA and extracellular metabolites is expected to improve the difficult economic balance of PHA production by maximizing the utilization of substrates. This feature is particularly relevant with regard to waste substrates of mixed composition.

## 2. Materials and Methods

### 2.1. Microorganism, Composition of Media, and Strain Storage

*Halomonas elongata* 1H9^T^ (DSM 2581) was purchased as a lyophilized culture from DSMZ—German Collection of Microorganisms and Cell Cultures GmbH. For storage purposes, *H. elongata* 1H9^T^ was cultivated in a modified HM medium [7]: (per liter) 45 g NaCl, 0.25 g MgSO_4_.7H_2_O, 0.5 g KCl, 5 g peptone, 10 g yeast extract, and 0.09 g CaCl_2_.2H_2_O. The strain was kept in cryovials at −80 °C with 15% (*w*/*v*) glycerol.

The medium for the seed culture had the following composition: (per liter) 45 g NaCl, 2.5 g MgSO_4_.7H_2_O, 2.5 g NH_4_Cl, 0.55 g K_2_HPO_4_, 1 mL trace elements [24], 8.9 g monosodium glutamate, 15 g Tris-HCl, and 20 g glucose. Solutions of MgSO_4_.7H_2_O and glucose, autoclaved separately, were supplemented to the sterile basal medium. The initial medium composition for the fed-batch cultures was (per liter): 45 g NaCl, 5 g MgSO_4_.7H_2_O, 12 g NH_4_Cl, 6 or 1 g K_2_HPO_4_ (under N or P-limiting conditions, respectively), 7.7 mL trace elements [24], 10 g monosodium glutamate, and 50 g glucose. The feed solution used in N- and P-limited cultivations had the following composition: (per liter) 45 g NaCl, 5 g MgSO_4_.7H_2_O, and 600 g glucose.

### 2.2. Fed-Batch Cultivations

Fed-batch cultivations were carried out in 2 L stirred-tank reactors (New Brunswick Bioflow 115) equipped with 2 six-bladed disk-turbine impellers, operated using BioCommand Batch Control software, which enables control, monitoring, and data acquisition. Bioreactors were inoculated with 10% (*v*/*v*) inoculum, reaching an initial working volume of 1.3 L of culture. The inoculum was prepared in duplicate (final volume 130 mL) by the inoculation of one cryovial into a 500 mL baffled Erlenmeyer containing 65 mL of seed culture at pH 8. Flasks were incubated in an orbital incubator (Infors AG, Bottmingen, Switzerland) at 35 °C, 200 rpm for 18 h.

Assays started with an initial optical density (OD_600_) around 0.5–0.8. Temperature and pH were maintained at 35 °C and 7.5, respectively. For pH control, NaOH (5–10 M) was used in fermentations under N-limitation, and NH_4_OH (30%) was used in fermentations under P- limitation. During the time course of the cultivations, the dissolved oxygen concentration (DOC) was set at 20% saturation. This value was maintained, while using a constant aeration rate of 2.6 L_air_/min, by controlling the stirring speed between 200 and 1200 rpm in cascade mode with DOC. The bioreactor was equipped with a foam sensor which allowed for the detection of foam and the automatic intermittent addition of an anti-foam solution (Simethicone Emulsion USP, Dow Corning) as needed. For the batch phase, the initial culture medium was designed to promote growth up to circa 20–25 g/L biomass. Thereafter, either nitrogen or phosphate was meant to become limited, promoting polymer accumulation. During the feeding stage, feed solution was supplemented by pre-programmed feeding pulses using the controlling software. Culture samples were periodically harvested to analyze biomass, polymer, sugar, nitrogen, and phosphate concentrations.

### 2.3. Analytical Methods

Cellular growth was monitored offline by measuring the optical density of culture samples at 600 nm with a double beam spectrophotometer (Hitachi U-2000). For cell dry weight analysis, 1.2 mL culture samples were centrifuged in a Sigma 1-15 P microcentrifuge (9168 g, 5 min) in pre-weighed and dried 1.5 mL microtubes. Cell pellets were then washed with distilled water, centrifuged for water removal, and dried at 62 °C in a Memmert GmbH oven (Model 400) until constant weight was achieved.

Sugar, organic acids, and phosphate concentrations were determined using high performance liquid chromatography (HPLC) (Hitachi LaChrom Elite- Hitachi High Technologies America, Inc., Schaumburg, IL, USA). This HPLC was equipped with a Rezex ROA-Organic acid H^+^ 8% (300 mm × 7.8 mm) column, an auto sampler (Hitachi LaChrom Elite L-2200), an HPLC pump (Hitachi LaChrom Elite L-2130), a Hitachi L-2490 refraction index detector (RI) for the detection of sugar and phosphate, and a Hitachi L-2420 UV-Vis detector for organic acids. A column heater for large columns (Croco-CIL 100-040-220P, 40 cm × 8 cm × 8 cm, 30–99 °C) was connected externally to the HPLC system, and the column was kept at 65 °C. A 5 mM solution of H_2_SO_4_ was used as the mobile phase at an elution rate of 0.5 mL/min. The injection volume was 20 μL. Solutions of glucose (Merck, Hong Kong, China), K_2_HPO_4_, and gluconic acid (Sigma-Aldrich, St. Louis, MO, USA) were used to prepare standard curves for each compound for subsequent peak detection and quantification in culture samples. Glucose concentration was also measured by a colorimetric method with a d-glucose assay kit (K-GLUC, Megazyme, Wicklow, Ireland). Ammonium concentration was determined using the phenate method described by Greenberg et al. [25]. Different methods for orthophosphate (PO_4_^3−^) determination were used, namely: (a) the spectrophotometric assay using standard test kits (LCK 350; manufacturer: Hach Lange, Düsseldorf, Germany) and a spectrophotometer (DR2800, Hach Lange); (b) the ascorbic acid method (4500-phosphorous) described by Greenberg et al. [25], and the above-described phosphate determination using the HPLC.

For P(3HB) quantification, 1.2 mL culture samples were taken and centrifuged in a Sigma 1-15 P microcentrifuge (9168 g, 5 min) to recover cell pellets. Cell pellets were subjected to acidic methanolysis as described before [26]. Samples of the organic phase were analyzed by gas chromatography using an Agilent Technologies 5890 series II gas chromatographer equipped with an FID detector and a 7683B injector. The capillary column was a HP-5 from Agilent J&W Scientific, 30 m in length and 0.32 mm in internal diameter. The oven, injector, and detector temperatures were kept constant at 60 °C, 120 °C, and 150 °C, respectively. Data acquisition and integration were performed by a Shimadzu CBM-102 Communication Bus Module and Shimadzu GC Solution software (Version 2.3), respectively. Peak identification was achieved using as standard 3-methyl hydroxybutyrate (Sigma). P(3HB) quantification was determined using a P(3HB) standard calibration curve.

Liquid chromatography–high resolution tandem mass spectrometry (LC–HRMS/MS)—was used for the identification and quantification of gluconic acid in the *H. elongata* 1H9T cultivation media. Aqueous solutions of the gluconic acid standard (Sigma) and samples of the glucose fermentation supernatants were analyzed on an ultra-high performance liquid chromatography system (UHPLC) Elute interfaced with a quadrupole-time-of-flight (QqTOF) Impact II mass spectrometer equipped with an electrospray ionization (ESI) source (Bruker Daltonics, Bremen, Germany). Chromatographic separation was carried out on a Luna Omega C18 Polar column (150 mm × 2.1 mm, 1.6 μm particle size; Phenomenex, Torrance, CA, USA) using a gradient elution of 0.1% formic acid in water (mobile phase A) and acetonitrile (mobile phase) at a flow rate of 270 µLmin^−1^ and at a controlled temperature of 35 °C. Details of the instrumentation parameters were previously described elsewhere [13].

## 3. Results

*Halomonas elongata* 1H9^T^ is a moderately halophilic bacterium able to produce P(3HB) under limitation of nitrogen or phosphorous and in the presence of an excess carbon source [6]. In this work, fed-batch cultivations using glucose as substrate were carried out using both strategies in order to assess the conditions leading to the highest polymer yields and productivities.

Figure 1 shows the evolution of residual biomass (*Xr*) and polymer [P(3HB)] concentrations, as well as total cell dry weight (CDW) and polymer cell content [P(3HB) (*w*/*w*%)] during the cultivation under N limitation. The residual biomass was calculated as the difference between cell dry weight and P(3HB) concentration and attained a value of approximately 20 g/L, as desired. The maximum production and accumulation of P(3HB) were 21.2 g/L and 53.0%, respectively. This polymer content is higher than values of 40–42% obtained by *H. elongata* spp. on glucose by other authors [4,6]. The maximum overall volumetric polymer productivity, maximum specific polymer productivity, and yield on glucose were 0.39 g_(P3HB)_/(L·h), 0.02 g(_P3HB)_/(g_Xr_.h), and 0.11 g _P(3HB)_/g _glucose_, respectively.

Additionally, assays were designed to attain P-limiting conditions. This is potentially a more effective PHA-producing strategy when using seaweed hydrolysates because of the high N content of the hydrolysate caused by the presence of algal proteins. To establish conditions for phosphate limitation, the concentration of the phosphate source (dibasic potassium phosphate) in the medium was lowered, and ammonium hydroxide was used as a base for pH control instead of sodium hydroxide in order to guarantee excess N nutrient. Different initial phosphate concentrations— namely 0.5, 1.0, 1.5 and 3.0 g/L K_2_HP0_4_—were tested. The highest polymer contents (33–35%) were obtained using 0.5 and 1 g/L K_2_HP0_4_. Results of the assay using 1.0 g/L K_2_HP0_4_ are represented in Figure 2. Under these conditions, the maximum P(3HB) produced was 20.0 g/L, corresponding to a P(3HB) accumulation of 33.3% (*w*/*w*). The overall volumetric polymer productivity g_P3HB_/(L.h), overall specific polymer productivity [g_P3HB_/(g_Xr_.h)], and the polymer yield on total glucose consumed were 0.33 g_P(3HB)_ /(L.h), 0.01 g _P3HB_/(g_Xr._h), and 0.08 g_P(3HB)_/g_glucose_, respectively. The low polymer content and low specific productivity obtained are ascribed to the high cell concentrations reached under these conditions. In fact, the cellular biomass (*Xr*) kept slightly increasing throughout the fermentation assay, including during the accumulation phase when the phosphate was expected to become exhausted. Different methods, mentioned in Section 2.3, were used to quantify and monitor phosphate concentration in the cultivation medium. Although correct phosphate levels were measured immediately prior to inoculation, phosphate levels decreased immediately to zero after addition of the inoculum.

Under nitrogen-limiting conditions, in addition to the fact that higher P(3HB) accumulations were reached, the HPLC analyses of the supernatants (with both UV-Vis and RI detectors) show excretion of an organic acid to the culture medium (Figure 1). Samples were examined by LC–HRMS, and the compound was identified as gluconic acid based on its *m*/*z* values and fragmentation pattern (Figure 3, Appendix A). Confirmation was made by analysis of literature data and comparison with the analysis of a gluconic acid standard. Another organic acid was also detected and identified as 2-oxoglutarate, an intermediate of the tricarboxylic acid cycle.

## 4. Discussion

Cultivations were designed for biomass to grow under nutrient-balanced conditions in order to attain a biomass concentration of circa 20 g/L, after which, upon exhaustion of one nutrient source (N or P), P(3HB) accumulation was promoted. During the time course of the cultivations, cell growth, glucose, ammonium or phosphate consumption, and polymer production were followed. The production and accumulation of organic acids were also monitored. The cultivations were optimized, and the results were compared. P(3HB) titers, volumetric productivities, and yield of product on sugar (Y_P(3HB)/glucose consumed_) were similar for both conditions, opposite to P(3HB) contents and specific productivities, which were much lower in the case of P-limitation. This is ascribed to a pronounced increase of the residual cell concentration (*Xr*) in the first 22 h of cultivation followed by a steady increase concurrent with polymer production. As phosphate levels in the media dropped to zero immediately after inoculation, it is suggested that *H. elongata* cells are able to accumulate polyphosphate reserves, leading to an increase in the residual cell concentration. Biomass growth (*Xr*) during the biopolymer accumulation phase in phosphate-limited fermentations has been observed before. Cultivations using *Burkholderia sacchari* and xylose as substrate showed an increase in biomass during the accumulation phase, even when phosphate was depleted in the cultivation medium [27]. The authors suggested the existence of polyphosphate reserves accumulated during the growth phase [27]. Bacteria are known to be able to accumulate inorganic phosphate under stress conditions such as nutritional, osmotic, or acidity stress. In fact, polyphosphate accumulation has been described before in *Halomonas* spp. [28,29]. Therefore, the inability to detect phosphate in the supernatant of the fermentation might thus be correlated with the capability of *H. elongata* 1H9^T^ to rapidly uptake the phosphate sources available in the medium for polyphosphate accumulation, which would then be used, as required, by the bacterial cells throughout the course of the cultivation. Scavenging of phosphate may even have started prior to the assay in the bioreactor, i.e., during inoculum growth.

Accumulation of gluconate and 2-oxoglutarate in the cultivation medium was detected in this work under N-limitation. Organic acid secretion and accumulation, namely of pyruvate and oxaloacetate, have already been described in wild *Halomonas* strains, such as *Halomonas* sp. KM-1 [16,17]. The production of gluconic acid by *Halomonas* spp. has been seldom reported. A work by Pastor et al., 2013 [18] registered the production of gluconic acid by *Chromohalobacter salexigens*, previously *H. elongata* 1H11. This is a halophilic and highly halotolerant bacterial strain able to grow in media containing up to 3M NaCl through the *de novo* synthesis of the compatible solute-ectoine. Those authors showed that the metabolism of this strain has adapted and is more efficient at high salinity (osmoadaptation). At low salinities, byproducts like gluconic acid, pyruvate, and acetic acid from overflow metabolism accumulate in the cultivation media. An excess of glucose in the media is not utilized by the central metabolism due to the lower activities of enzymes at low salt concentrations caused by the reduced demand for ectoines. In the case of glucose, it is diverted to the production of gluconic acid (Figure 4). These overflow metabolites are later re-assimilated.

Recently, a work published by our group with *Halomonas boliviensis* [13] has also reported the production of gluconic acid.

The maximum gluconic acid concentration attained with *H. elongata* 1H9^T^ was circa 133 g/L (Figure 1) which, compared to the production levels attained by other gluconic-producing strains [25], indicates that this strain is a promising gluconic acid producer. As an example, strains of *Aspergillus niger* were reported to produce gluconic acid titers from 54 to 76 g/L using complex substrates such as fruit must/molasses or sugarcane molasses [30,31]. Also, titers of gluconic acid produced by *Gluconobacter oxydans* varied between 69 and 90.3 g/L depending on the pH of the culture medium [32]. Gluconic acid is a valuable organic acid that finds extensive application in a variety of areas such as in the food, pharmaceutical, chemical, and construction industries. Therefore, there is a high demand for designing more efficient and low-cost processes for gluconic acid production. Currently, the fungus *Aspergillus niger* and the acetic acid bacterium *Gluconobacter oxidans* [31] are commonly used by industry for the biological production of gluconic acid.

In the present study, a fraction of glucose initially meant for P(3HB) accumulation is being diverted into gluconic acid production, lowering P(3HB) productivity and yield. However, the concomitant production of gluconic acid, with high commercial value, might drive the co-process to economic feasibility. This strategy was already suggested for the co-production of P(3HB) and the valuable compatible solute ectoine by *Halomonas* spp. [6,33]. On the other hand, if gluconic acid is the focus metabolite, the metabolic pathways involved in its production must be known. The metabolic pathways that generally are involved in glucose utilization are the Embden–Meyerhof–Parnas (EMP) for energy generation, the gluconate pathway, the pentose phosphate pathway (PPP) for reducing power (NADPH), fatty acids and nucleotide biosynthesis, and the Entner–Doudoroff (ED) pathway. In *Halomonas elongata*, the glucose metabolism has been suggested to be conducted through the Entner–Doudoroff (ED) pathway [34] (Figure 4).

For *H. elongata* 1H9^T^ a periplasmic variant of the ED pathway has been proposed involving the periplasmic oxidation of *d*-glucose into *d*-gluconate through the action of a pyrroloquinoline-quinone-dependent glucose dehydrogenase found encoded in the genome. Gluconate would then enter the cell by a hypothetical transporter, and, in the cytoplasm, through the action of a glucokinase (also found encoded in the genome), would be converted into 6-phosphogluconate, an intermediate in the ED and PPP pathways [34]. Such a gluconate shunt could act, as described before, as a way to direct glucose and gluconate into the ED pathwaybypassing the rate-limiting step of the PPP [35]. It is however worth recalling that the regeneration of reduced power though the PPP is critical for P(3HB) production. During the formation of P(3HB), two molecules of acetyl-CoA are condensed, resulting in a molecule of acetoacetyl-CoA, which is reduced by NADPH-dependent acetoacetyl-CoA reductase to (*R*)-3-hydroxybutyryl-CoA, which, in turn, is subsequently polymerized by the action of a PHB synthase [5]. From our studies, we observed that the production of gluconic acid occurred with simultaneous glucose utilization when glucose levels in the culture were high (Figure 1). When glucose levels significantly diminished (Figure 1b) in the culture medium between feeding pulses, as seen between 32 h and 46 h (Figure 1c), gluconic levels also decreased from 133 g/L to 27 g/L (Figure 1b), suggesting its utilization by the cell as a carbon source. When glucose levels were replenished with glucose feed pulses at 45 h–50 h (Figure 1b,c), gluconate started to accumulate again in the culture medium, reaching 64 g/L at 55 h. The concentration of gluconic acid subsequently decreased, as no additional glucose feeding pulses were provided until the end of the assay following the last pulse given at around 50 h (Figure 1c). These observations suggest that when glucose levels are high, oxidation of glucose into gluconate occurs in the periplasm and reducing power is renewed. However, under such conditions, it appears that excess gluconate is secreted into the bulk medium. Then, when glucose levels decrease, a hypothetical control mechanism might be activated, allowing gluconate to be channeled from the culture medium into the cell in order to be used as a carbon source. Further studies are necessary to elucidate the molecular mechanisms subjacent to the observed glucose metabolism.

From the above, it becomes clear that a co-production process of gluconic acid and P(3HB) needs to be designed in order to ensure high glucose levels throughout the course of the bacterial cultivation and to promote the production of both products. For their recovery at the end of the cultivation, separation of the P3HB-containing biomass from the medium containing gluconic acid should take place. P(3HB) recovery from cell biomass can be accomplished by solvent extraction followed by the use of an anti-solvent for polymer recovery [36] or through the lysis of the cell membrane using aqueous solutions of NaOH, HClO, or surfactants [37]. As for the method of gluconic acid recovery from the culture media, application of established protocols would need to be further optimized keeping in mind the saline conditions of the fermentation medium [31].

Additionally, it is important to note that under conditions of phosphate limitation, gluconic acid accumulation has not been observed. This might indicate that the phosphorous availability in the medium plays a role in the pathways that conduct to gluconic acid formation.

## 5. Conclusions

Fed-batch cultivations with *H. elongata* 1H9^T^ and glucose as substrate attained similar P(3HB) titers, overall volumetric productivities, and yields of P(3HB) on glucose using either nitrogen or phosphate limitation conditions, although N-limited cultivations performed slightly better. Under nitrogen limitation, a P(3HB) accumulation of 53.0%, a yield on glucose of 0.11 g_P(3HB)_/g_glucose_, and an overall productivity of 0.39 g_P(3HB_/(L·h) were reached. Furthermore, *H. elongata* 1H9^T^ has been shown to be an excellent gluconic acid producer. Cultivations under nitrogen limitation led to the co-production of P(3HB) and 133 g/L gluconic acid, a value-added product. The production of this organic acid appears to be related to high glucose concentrations in the medium. The design of fed-batch fermentation processes maintaining high glucose concentrations in the culture medium and nitrogen limitation conditions is a promising strategy to promote P(3HB) accumulation in the cell’s cytoplasm and, simultaneously, gluconic acid production. The findings described in this work are relevant though preliminary, and, as such, this co-production process calls for future attention. Further studies are required to better understand the molecular mechanisms involved in gluconic acid production by this strain, the roles of glucose and oxygen concentrations, nitrogen limitation, and other cultivation parameters which favor a balance between P(3HB) and gluconic acid production.

## Figures and Tables

**Figure 1 bioengineering-10-00643-f001:**
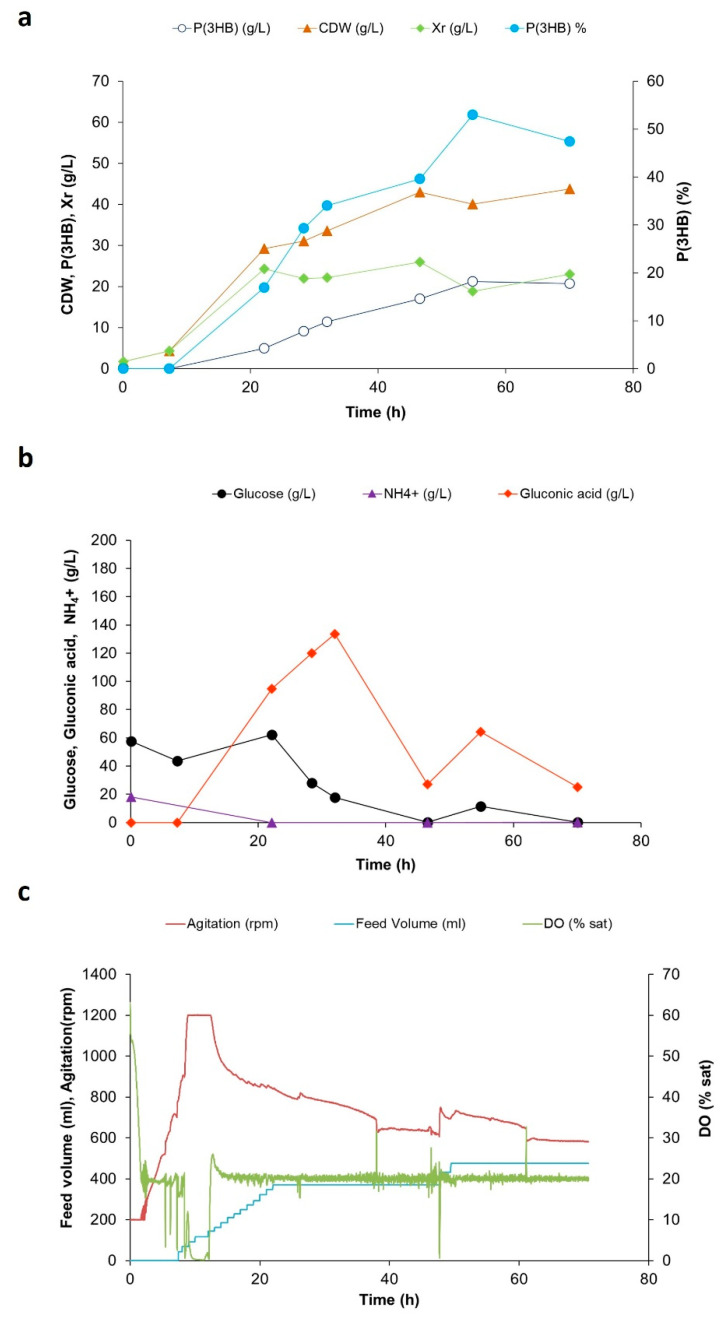
Fed-batch culture of *Halomonas elongata* 1H9^T^ with glucose as a carbon source and under nitrogen limiting conditions. Cell biomass and P(3HB) production (**a**), glucose and ammonium utilization and gluconic acid production (**b**), and data acquired automatically during the cultivation, namely feed volume, agitation, and DO (% sat), (**c**).

**Figure 2 bioengineering-10-00643-f002:**
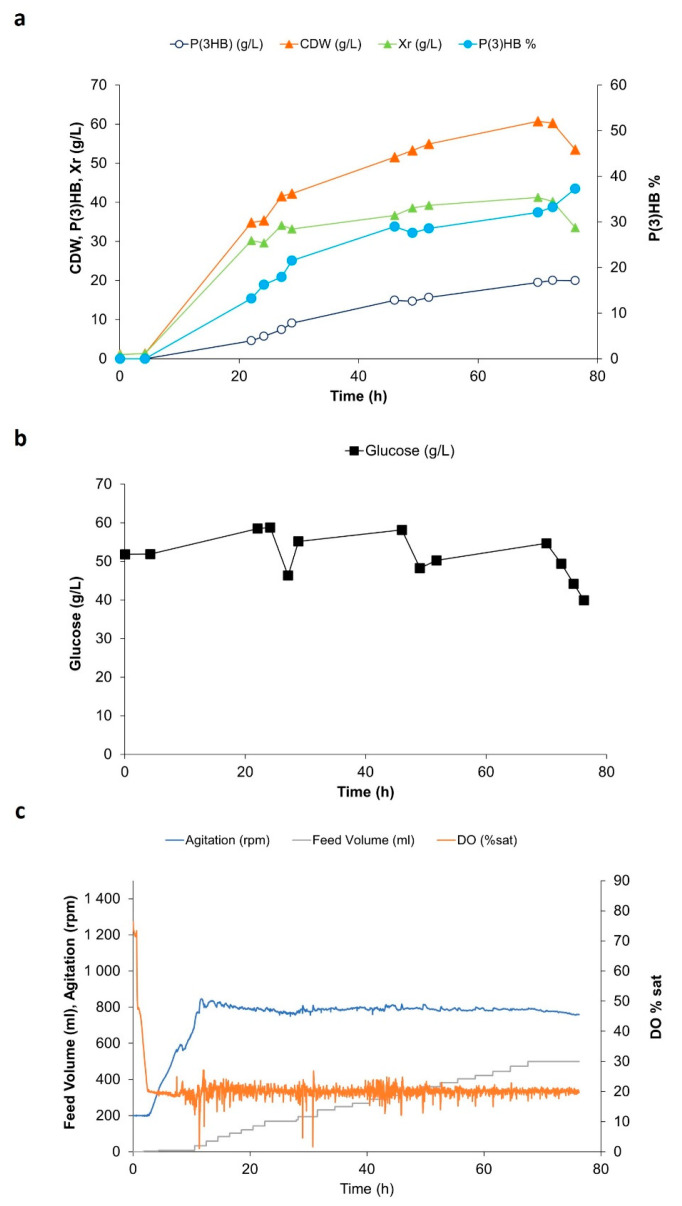
Fed-batch culture of *Halomonas elongata* 1H9^T^ with glucose as a carbon source and under phosphate limiting conditions. Cell biomass and P(3HB) production (**a**); sugar utilization (**b**); feed volume, agitation, and DO% sat (**c**).

**Figure 3 bioengineering-10-00643-f003:**
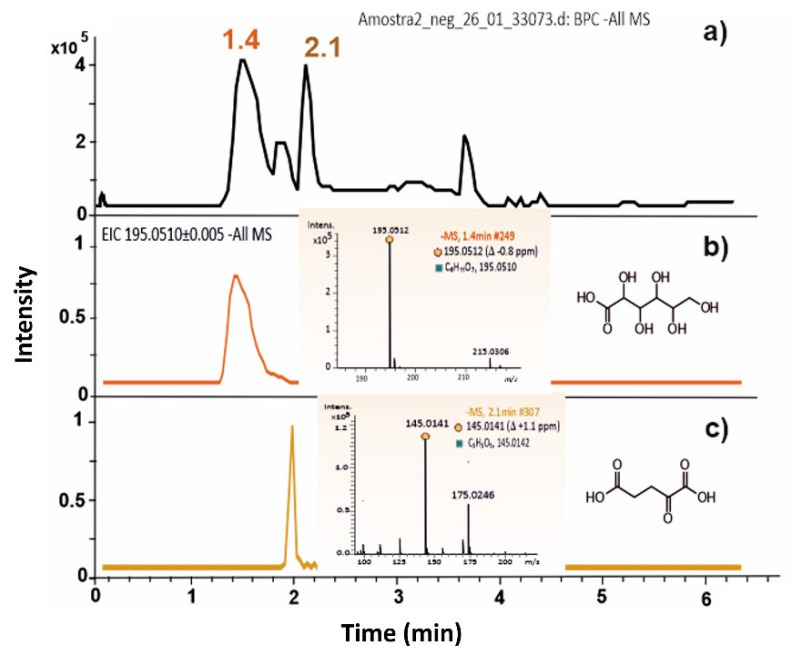
LC–HRMS analysis of a sample of the cultivation broth of *Halomonas elongata* 1H9^T^ on glucose under N-limiting conditions. (**a**) Total ion chromatogram acquired in the ESI negative mode. Two main extracted ion chromatograms were obtained: (**b**) for the deprotonated molecule of gluconic acid [C6H11O7]^−^, tR 1.4 min, *m*/*z* 195.0509 (∆ −0.5 ppm, mSigma 5.5); and (**c**) for the deprotonated molecule of 2-oxoglutaric acid [C5H5O5]^-^, tR 1.9 min, *m*/*z* 145.0148 (∆ −3.5 ppm, mSigma 8.5). Each precursor ion was selected by the quadrupole mass analyzer, transferred to the collision cell where fragmentation occurred by collision-induced dissociations (CID) with nitrogen, and the fragmented ions were separated by the TOF mass analyzer, yielding the high-resolution tandem mass spectra. The assignment of the fragmented ions was based on their accurate mass measurements, and the fragmentation paths are proposed in Appendix A.

**Figure 4 bioengineering-10-00643-f004:**
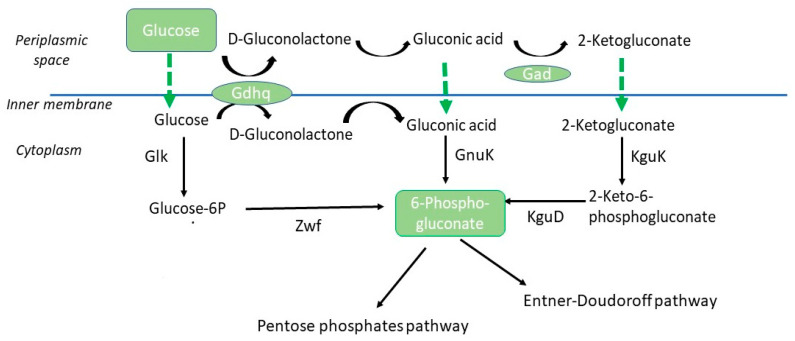
Pathways leading from glucose to 6-phosphogluconate in *Halomonas elongata* 1H9T. Abbreviations used are as follows: Gad, gluconate dehydrogenase; Gdhq, glucose dehydrogenase; Glk, glucokinase; GnuK, gluconokinase; KguD, 2-keto-6-phosphogluconate reductase; KguK, 2-ketogluconatekinase; Zwf, glucose-6-phosphate dehydrogenase.

## Data Availability

All data are available in the paper.

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
