# Peer review of "Co-Production of Poly(3-hydroxybutyrate) and Gluconic Acid from Glucose by Halomonas elongata"

_bioengineering, 2023, doi:10.3390/bioengineering10060643_

Round 1
Reviewer 1 Report
1) Co-production is mentioned on line 24 but its current status in connection with polyhydroxyalkanoate synthesis is not addressed in the introduction. This should be remedied by including review articles on the subject which will illustrate the wide range of value-added products being studied.
2) Related to the above is the fact that gluconic acid can be produced from glucose by a number of synthesis routes. This raises the question of the economic competitiveness of the work described in the manuscript when compared with other reported substrates and value-added co-products, such as hydrogen. A short analysis should be included in the discussion especially from the standpoint that the world needs to increase energy resources and maybe can get by without food additives and skin care products.
3) It is not clear what the glucose concentration levels in the feed pulses described on line 329 at 45 and 50h were, for example a return to the initial 600 g/L stated on line 88. If this is the case does that mean that the gluconic acid concentration only increases to about 60 g/L after the 50h addition because N is starting to run out?
4) It is stated on line 41 that PHA production occurs at nutrient-limiting and carbon excess conditions. The experimental procedure to demonstrate this is somewhat confusing.
Lines 85-89 indicate the molar quantities involved: 3.3 glucose, 0.22 ammonium chloride and 0.01 (from line 203) dipotassium hydrogen phosphate. If it were a chemical reaction being studied, excess glucose concentration would be constant, while the limiting N and P reactants would be rate determining. Cell metabolism doesn't follow such guidelines and as can be seen in Figure 1b the glucose reagent in excess goes to zero at about 45h, approximately 25h after the limiting reagent N was no longer detected in the medium. In addition, Figure 2b shows a constant concentration of glucose substrate while product PHA continually increases. A clarification is needed of these observations.
There is a significant difference in the concentrations of the so-called limiting reagents. P with the lower concentration is described as never being present in the medium on line 253. But N with higher initial concentration also reaches zero in Figure 1b but is never described as being accumulated in cells, as P is on line 254, despite the continuing production of PHA after 22h shown in 1a. A clarification is needed.
It is not clear in section 2.1 what concentrations of P and N nutrients were when they were non-limiting nutrients. Curves should be added for phosphate in Figure 1b and ammonium in Figure 2b or at least a description of their trends, for example constant throughout, to the text.
5) There are several control experiments which should either be considered or dismissed as not being relevant.
Ectoine production is described as being in response to osmotic imbalances in reference 6. It was never observed as a product, presumably because of the relatively low salinity level employed. Of interest would be the NaCl level where ectoine production would become measurable at the given conditions.
The product gluconic acid is said to reenter the cell as a carbon source on line 337. This seems to imply a different product distribution at that condition since it is unlikely that gluconic acid as substrate would continue to produce gluconic acid. A control experiment using gluconic acid as the initial substrate is needed in order to confirm the hypothesis.
Results are presented for cultures with either N or P limiting conditions. The prospects for gluconic acid production, at the expense of PHA, when both are non-limiting would be of interest.
6) No fragmentation patterns are shown in the supplementary material.
Author Response
1) Co-production is mentioned on line 24 but its current status in connection with polyhydroxyalkanoate synthesis is not addressed in the introduction. This should be remedied by including review articles on the subject which will illustrate the wide range of value-added products being studied.
Thanks for the note. We have added a new paragraph with references of three review articles on co-production and respective discussion (lines 74-89 in the revised manuscript).
2) Related to the above is the fact that gluconic acid can be produced from glucose by a number of synthesis routes. This raises the question of the economic competitiveness of the work described in the manuscript when compared with other reported substrates and value-added co-products, such as hydrogen. A short analysis should be included in the discussion especially from the standpoint that the world needs to increase energy resources and maybe can get by without food additives and skin care products.
The above paragraph also addresses this issue. We agree that the production of biofuels is of the outmost importance, but we think the production of organic acids by a biorefinery approach is also relevant. Moreover, Halomonas spp are not hydrogen producers.
3) It is not clear what the glucose concentration levels in the feed pulses described on line 329 at 45 and 50h were, for example a return to the initial 600 g/L stated on line 88. If this is the case does that mean that the gluconic acid concentration only increases to about 60 g/L after the 50h addition because N is starting to run out?
As reported in Figure 1b, the initial nitrogen source was exhausted in the first 24h and remained limited throughout the assay. We only observed production of P3HB and gluconic acid once N limiting conditions were established. The feed solution had in its composition 600 g/L glucose. Feeding pulses were added in an attempt to partially restore glucose levels in the bulk medium. However, in practice, glucose levels fluctuated as glucose was being used versus what we provided with the feed pulses (Fig 1b).
In lines 343-348 (in new revised document) what we intend to convey was the observation that following a feed pulse, gluconic acid was secreted, but, with continued glucose uptake, its concentration subsequentially dropped and the culture appeared to use gluconic acid as carbon source. As discussed, further studies are necessary to evaluate this feedback control process responsible for the observations reported.
4) It is stated on line 41 that PHA production occurs at nutrient-limiting and carbon excess conditions. The experimental procedure to demonstrate this is somewhat confusing.
Lines 85-89 indicate the molar quantities involved: 3.3 glucose, 0.22 ammonium chloride and 0.01 (from line 203) dipotassium hydrogen phosphate. If it were a chemical reaction being studied, excess glucose concentration would be constant, while the limiting N and P reactants would be rate determining. Cell metabolism doesn't follow such guidelines and as can be seen in Figure 1b the glucose reagent in excess goes to zero at about 45h, approximately 25h after the limiting reagent N was no longer detected in the medium. In addition, Figure 2b shows a constant concentration of glucose substrate while product PHA continually increases. A clarification is needed of these observations.
Glucose concentration was not constant throughout the assay. It varied as the bacteria consumes glucose and as we provide feed pulses to replenish glucose again. At 45h the concentration reached 0 g/L, however it rises again after that as we provide another feed pulse. Feed addition throughout the assay is depicted in Figure 1c.
There is a significant difference in the concentrations of the so-called limiting reagents. P with the lower concentration is described as never being present in the medium on line 253. But N with higher initial concentration also reaches zero in Figure 1b but is never described as being accumulated in cells, as P is on line 254, despite the continuing production of PHA after 22h shown in 1a. A clarification is needed.
We describe in this work two different sets of experiments. One experimental condition for which results are presented in Figure 1 reports on the results obtained under N limiting conditions in the culture medium. Here the culture medium had an initial source of N in the form of NH4Cl. This N source was used by the cells to grow until reaching a biomass of around 20 g/L which happens within the first 22h of the assay. From that point on, growth ceases and under N-limiting conditions the biomass present starts to produce P3HB and to produce gluconic acid which is secreted into the culture medium as observed in Figure 1b.
The second experimental condition, for which the results are present in Figure 2, reports on the results obtained under P limiting conditions. We start with a lower P concentration as compared to the previous assay (1 g K2HPO4) so as to promote initial growth up to around 20 g/L biomass. We report in line 269 (line numbering in the new revised manuscript) that P levels dropped to zero following inoculation and we suggested that the bacteria store this initial P as reserves which they will manage throughout the assay. We thus proposed, similarly to what has been described before in the literature, that the extent of bacterial growth observed during the biopolymer accumulation phase might be explained by a phosphorous scavenging process taking place shortly after the inoculation time.
It is not clear in section 2.1 what concentrations of P and N nutrients were when they were non-limiting nutrients. Curves should be added for phosphate in Figure 1b and ammonium in Figure 2b or at least a description of their trends, for example constant throughout, to the text.
As discussed in line 227 (line numbering in the new revised manuscript) different methods, mentioned in section 2.3., were used to quantify and monitor phosphate concentration in the cultivation medium. Although correct (i.e., consistent with the medium recipe) phosphate levels were measured immediately prior inoculation, phosphate levels decreased immediately to zero after addition of the inoculum. In lines 269-284 (line numbering in the new revised manuscript) we propose that H. elongata cells are able to accumulate P as polyphosphate reserves leading to the observed increase of the residual cell concentration throughout the assay. Biomass growth (Xr) during the biopolymer accumulation phase in phosphate-limited fermentations has been observed before. In the work by Oliveira-Filho et al (2020), using Burkholderia sacchari and xylose as substrate, the occurrence of an increase in biomass during the accumulation phase was observed, even when phosphate was depleted in the cultivation medium. Likewise, the authors suggested the existence of polyphosphate reserves accumulated during the growth phase.
In our experimental design we considered that monitorization of N levels was only relevant for experimental assays performed under N limiting conditions. Under P limiting conditions, N was never limited since for pH control a solution of NH4OH(30%) was used for automatic pH control.
5) There are several control experiments which should either be considered or dismissed as not being relevant.
Ectoine production is described as being in response to osmotic imbalances in reference 6. It was never observed as a product, presumably because of the relatively low salinity level employed. Of interest would be the NaCl level where ectoine production would become measurable at the given conditions.
The product gluconic acid is said to reenter the cell as a carbon source on line 337. This seems to imply a different product distribution at that condition since it is unlikely that gluconic acid as substrate would continue to produce gluconic acid. A control experiment using gluconic acid as the initial substrate is needed in order to confirm the hypothesis.
Results are presented for cultures with either N or P limiting conditions. The prospects for gluconic acid production, at the expense of PHA, when both are non-limiting would be of interest.
Ectoine production is indeed a relevant compatible solute produced by Halomonas spp. Some Halomonas strains can produce ectoine at low salt conditions. However, levels of ectoine production were not the subject of evaluation of this study. We fully agree that in future studies, monitorization of such product is a relevant factor to account for a better understanding of H. elongata metabolism.
As described, we propose that gluconic acid is being used as an alternative C source when glucose levels are low, thus reentering the cell. It will be interesting, as suggested, to evaluate several metabolic parameters in conditions using gluconic acid as sole carbon source.
Likewise, such assay as suggested with non-limitation of both N and P may also further elucidate the role of each nutrient limitation on co-production of gluconic acid and P3HB.
6) No fragmentation patterns are shown in the supplementary material.
Fragmentation patterns are provided in supplementary materials Figure 1S and 2S (line 239).
Reviewer 2 Report
Using the Halomonas elongate strain, preliminary results showed a promising gluconic acid producer.
As the authors themselves suggested, additional testing of the production conditions of gluconic acid is needed.
Author Response
Using the Halomonas elongata strain, preliminary results showed a promising gluconic acid producer.
As the authors themselves suggested, additional testing of the production conditions of gluconic acid is needed.
R: The assessment and optimization of the gluconic acid production by this strain is a study programmed in a very near future.
Reviewer 3 Report
In this contribution, the authors describe the production of poly-3-hydroxybutyrate (P(3HB)) form glucose by Halomonas elongata, both under N- and P-deficient conditions. Regardless of the conditions, the productivity of P(3HB) was comparable, albeit slightly higher levels were attained in the N-deficient cultures. Along with that of P(3HB), the production of gluconic acid was also observed under nitrogen-limited conditions.
The draft is well written, nicely presented and each section is clearly described. I believe this work would be of interest for the readers of Bioengineering, I hence recommend its publication after addressing some (minor) comments.
1) Given the preliminary (albeit valuable) nature of the results, I wonder if the present manuscript would be better described as a Communication, rather than as a Full Paper.
2) Figures 1a and 2a: along with color coding, please consider using different line styles or markers to differentiate P(3HB) % and P(BH3) g/L datasets so that the chart could be easily be read in black and white.
3) For the paragraph starting at line 285: could the authors please provide some values for the gluconic acid production levels by other strain, so that the comparison with the present investigation would be more prompt.
Author Response
In this contribution, the authors describe the production of poly-3-hydroxybutyrate (P(3HB)) form glucose by Halomonas elongata, both under N- and P-deficient conditions. Regardless of the conditions, the productivity of P(3HB) was comparable, albeit slightly higher levels were attained in the N-deficient cultures. Along with that of P(3HB), the production of gluconic acid was also observed under nitrogen-limited conditions.
The draft is well written, nicely presented and each section is clearly described. I believe this work would be of interest for the readers of Bioengineering, I hence recommend its publication after addressing some (minor) comments.
1) Given the preliminary (albeit valuable) nature of the results, I wonder if the present manuscript would be better described as a Communication, rather than as a Full Paper.
We would prefer to keep it as Full Paper, if acceptable.
2) Figures 1a and 2a: along with color coding, please consider using different line styles or markers to differentiate P(3HB) % and P(BH3) g/L datasets so that the chart could be easily be read in black and white.
As suggested new Figures are provided with a novel line style for improved visualization.
3) For the paragraph starting at line 285: could the authors please provide some values for the gluconic acid production levels by other strain, so that the comparison with the present investigation would be more prompt.
As suggested a new paragraph was introduced starting at line 304 in the new revised manuscript for comparison of our results with the literature.
Round 2
Reviewer 1 Report
Still wary of the terminology on line 125 that the nutrients are limiting and promoting at the same time, there ought to be some cell death involved, but not my field.